# Laser Micropatterning Promotes Rete Ridge Formation and Enhanced Engineered Skin Strength without Increased Inflammation

**DOI:** 10.3390/bioengineering10070861

**Published:** 2023-07-20

**Authors:** Britani N. Blackstone, Megan M. Malara, Molly E. Baumann, Kevin L. McFarland, Dorothy M. Supp, Heather M. Powell

**Affiliations:** 1Department of Materials Science and Engineering, The Ohio State University, 140 W 19th Avenue, Columbus, OH 43210, USA; 2Department of Biomedical Engineering, The Ohio State University, 140 W 19th Avenue, Columbus, OH 43210, USA; 3Department of Surgery, University of Cincinnati College of Medicine, 231 Albert Sabin Way, Cincinnati, OH 45267, USA; 4Center for Stem Cell & Organoid Medicine (CuSTOM), Cincinnati Children’s Hospital Medical Center, 3333 Burnet Avenue, Cincinnati, OH 45229, USA; 5Shriners Children’s Ohio, 1 Children’s Plaza, Dayton, OH 45404, USA

**Keywords:** scaffold, laser ablation, rete ridge, engineered skin, inflammation, biomechanics

## Abstract

Rete ridges play multiple important roles in native skin tissue function, including enhancing skin strength, but they are largely absent from engineered tissue models and skin substitutes. Laser micropatterning of fibroblast-containing dermal templates prior to seeding of keratinocytes was shown to facilitate rete ridge development in engineered skin (ES) both in vitro and in vivo. However, it is unknown whether rete ridge development results exclusively from the microarchitectural features formed by ablative processing or whether laser treatment causes an inflammatory response that contributes to rete ridge formation. In this study, laser-micropatterned and non-laser- treated ES grafts were developed and assessed during culture and for four weeks post grafting onto full-thickness wounds in immunodeficient mice. Decreases in inflammatory cytokine secretion were initially observed in vitro in laser-treated grafts compared to non-treated controls, although cytokine levels were similar in both groups five days after laser treatment. Post grafting, rete ridge-containing ES showed a significant increase in vascularization at week 2, and in collagen deposition and biomechanics at weeks 2 and 4, compared with controls. No differences in inflammatory cytokine expression after grafting were observed between groups. The results suggest that laser micropatterning of ES to create rete ridges improves the mechanical properties of healed skin grafts without increasing inflammation.

## 1. Introduction

Rete ridges are epithelial penetrations into the stroma found in native skin, oral mucosa, and the cornea that play major roles in both tissue homeostasis and wound healing. Current tissue engineering methods have begun to investigate the relationship between rete ridge architecture and cell behaviors [1,2,3,4,5,6], including proliferation, differentiation, and stem cell patterning, and to incorporate these important features into cultured tissue substitutes [7,8,9,10]. Rete ridges can be fabricated by casting onto molds [1,2,3,4,8,9,10,11,12,13], electrospinning onto molds [6] or patterned collectors [7] that have been manufactured with these features, using air pressure to create a dynamic culture substrate with these features [5], and bioprinting [14]. While the desired architecture can be achieved using these methods, some methods have significant limitations related to how well the structure is preserved with culture time [12] or whether keratinocytes and fibroblasts are in direct contact [8,11]. Direct contact between keratinocytes and fibroblasts facilitates paracrine regulation by reducing diffusion distances for soluble factors and allowing more direct mechanical interactions, similar to epidermal and dermal layers in uninjured skin. The ability to form rete ridges in a model where the epidermis and dermis are in direct contact would allow for the study of epidermal–dermal communication in epidermal attachment and skin development and could bring the biomechanics of engineered skin closer to those of native skin.

Laser ablation can be used to modify scaffold topography [15,16] in a wide array of ablative well shapes, depths, and spacing by tuning laser power, density, and pattern. Recent studies developing rete ridges in engineered skin [17] and in grafted cultured epithelial autografts [18] utilized fractional carbon dioxide (FXCO_2_) laser ablation to micropattern fibroblast-seeded electrospun collagen-based dermal templates. The micropatterned wells promoted rete ridge formation and subsequently improved barrier function, epidermal proliferation, and basement membrane development compared to grafts with a flat interface at the dermal–epidermal junction (DEJ) [17]. Keratinocytes are known to be responsive to the morphology of their environment [1,2,4,19,20] and it was hypothesized that the physical structures of wells are directly responsible for the generation of rete ridge structures in engineered skin. However, it is not clear whether the laser technique used to micropattern the dermal template results in an inflammatory response that may promote or hinder the ridge formation process. Fractional CO_2_ ablation of skin and scar tissues has been shown to alter the expression of heat shock proteins (HSPs), transforming growth factor β (TGF-β), matrix metalloproteinases (MMPs), and other factors in a time-dependent manner [21]. As these factors are related to tissue development and remodeling, it is possible that laser-induced molecular changes in the dermal template may be as important as the microarchitectural features in directing the formation of rete ridges.

FXCO_2_-induced inflammation has been tracked up to seven days post lasering with an FXCO_2_ laser in human and porcine studies [22,23,24]. Increases in inflammatory cytokines interleukin-1β (IL-1β), interleukin-6 (IL-6), and monocyte chemoattractant protein-1 (MCP-1) were seen immediately following laser treatment and tapered off after four days [22,23]. TGF-β was upregulated modestly at early time points and was maintained through seven days [22], while vascular endothelial growth factor (VEGF) was elevated one day post lasering and peaked after three days compared to uninjured skin [24]. Thus, it is possible that these factors may also be upregulated by FXCO_2_ laser ablation in an engineered dermal/skin model.

Ablative lasers used clinically for resurfacing and matrix remodeling [25,26] have been used in other in vitro engineered skin models. A fractional erbium:YAG laser was used by Schmitt et al. to study the molecular effects of ablative laser treatment in a cultured skin equivalent [27]. Three days after laser treatment, increased mRNA expression was found for inflammatory cytokines IL-6 and IL-8, and for multiple MMPs responsible for collagen degradation after ablative lasering [27]. In another study, the effect of FXCO_2_ laser ablation was investigated in a full-thickness cultured skin model five days after the seeding of keratinocytes [28]. Five days post laser treatment with an energy density of 100 mJ/cm^2^, genes associated with the immune response, tissue remodeling and wound healing, and members of the HSP family were upregulated, and MMPs and differentiation markers were downregulated [28]. While these full-thickness skin substitutes were used as in vitro models to investigate the molecular effects of clinical ablative laser use, of interest to the current study is the response to ablative laser treatment of a dermal component, with subsequent keratinocyte seeding to form engineered skin with rete ridges.

The goal of this study was to probe the in vitro inflammatory response to laser patterning and the downstream tissue development and biomechanics post grafting. In this study, dermal templates were developed with human dermal fibroblasts cultured on electrospun collagen scaffolds. Templates were patterned with an ablative FXCO_2_ laser to create dermal papillae-like structures and then seeded with human keratinocytes. Samples were assessed over a seven-day period to track inflammatory responses compared to non-lasered controls. Epidermal barrier function, mechanics, collagen composition, vascularization, and gene expression of inflammatory markers were then assessed 2 and 4 weeks after grafting to immunodeficient mice.

## 2. Materials and Methods

### 2.1. Fabrication of Engineered Skin

Scaffolds for the engineered skin grafts were constructed and prepared for cell seeding as previously described [17]. Briefly, scaffolds were electrospun from a solution of bovine collagen type I (DSM Biomedical, Exton, PA, USA) solubilized in hexafluoroisopropanol (HFP; Oakwood Chemical, Estill, SC, USA). Scaffolds were physically and chemically cross-linked, disinfected in 70% ethanol, and rinsed extensively with phosphate- buffered saline, followed by HEPES-buffered saline, and finally cell culture medium.

Engineered skin (ES) grafts with or without rete ridges were developed and cultured as previously described [17] (Figure 1), with fibroblasts and keratinocytes isolated from de-identified human skin acquired from the breast reduction of a 28-year-old African American female. This activity was reviewed by the Institutional Review Boards of Ohio State University and the University of Cincinnati and was determined not to involve human subjects research, therefore patient consent for use of de-identified tissue was not required. Cells were isolated and cultured in selective medium for each cell type, as detailed elsewhere [17]. Briefly, the epidermis was enzymatically dissociated from the dermis using Dispase II (Thermo Fisher Scientific, Waltham, MA, USA; Cat# 04942078001). Subsequently, keratinocytes were released from epidermal pieces using Trypsin digestion (Sigma-Aldrich, St. Louis, MO, USA; Cat# T8003), and fibroblasts were released from dermal tissue pieces using Collagenase Type I (Worthington Biochemical Corp., Lakewood, NJ, USA; Cat# LS004196) as detailed elsewhere [17]. Visual inspection of isolated cells and staining for specific markers, including vimentin for fibroblasts and keratin 14 for keratinocytes, confirmed cellular identities. Cells at passage 2 were used for all experiments.

Primary human dermal fibroblasts (HF) were seeded onto collagen scaffolds at 5 × 10^5^ cells/cm^2^ and cultured for five days, followed by seeding of epidermal keratinocytes (HK) at 1 × 10^6^ cells/cm^2^. For Ridged samples, an Ultrapulse^®^ fractional carbon dioxide laser (FXCO_2_, DeepFX^TM^ handset, Lumenis Inc., San Jose, CA, USA) was used to micropattern the dermal construct with wells approximately 80 µm in depth and 540 µm in width [17] prior to keratinocyte seeding. ES constructs were lifted to the air-liquid interface 1 day after HK inoculation and cultured an additional 9–10 days until grafting. ES culture medium was comprised of DME (Sigma-Aldrich) supplemented with 5 µg/mL bovine insulin (Sigma-Aldrich; Cat# I6634), 0.1 mM ascorbic acid-2-phosphate (Sigma-Aldrich; Cat# A8960), 0.5 µg/mL hydrocortisone (Sigma-Aldrich; Cat# H4001), 20 pM triiodothyronine (Sigma-Aldrich; Cat# T5516), 2 µg/mL linoleic acid (Sigma-Aldrich; Cat# L1012), 1 mM strontium chloride (Sigma-Aldrich; Cat# 439665), and 1x antibiotic-antimycotic solution (Thermo Fisher Scientific; Cat# 15-240-096). Progesterone (0.76 nM; Sigma-Aldrich; Cat# P8783) and EGF (10 ng/mL; Thermo Fisher Scientific; Cat# AF-100-15) were also added for the first three days of culture but were subsequently removed to promote epidermal differentiation [17].

### 2.2. Analysis of the In Vitro Inflammatory Response to Laser Patterning

Equal-size samples of dermal templates for analysis of inflammatory cytokine secretion were prepared following fibroblast seeding using 10 mm biopsy punches. These dermal template samples were randomly assigned into Flat (non-laser patterned) or Ridged (laser patterned) groups. After laser treatment of samples in the Ridged group, and HK seeding and attachment, samples were placed on N-terface^®^ (Winfield Laboratories, Inc., Richardson, TX, USA), a thin, non-adherent netting, and were floated at the air–liquid interface. Samples were cultured in individual wells of a 6-well plate and received daily medium changes.

Samples and culture medium were collected for analysis at 6 h, day 1, day 3, day 5, and day 7 post laser treatment. Note that samples were incubated at the air–liquid interface for 3 h after seeding HK to facilitate attachment prior to submersion in medium, so for 6 h and day 1 time points, samples were incubated in the collected medium for 3 h and 21 h, respectively. At all other time points, samples were incubated in the collected medium for 24 h prior to collection. Samples and culture medium were snap frozen separately in liquid nitrogen and stored at −80 °C until analyzed (*n* = 5 per time point and group). Levels of secreted factors were quantified by enzyme-linked immunosorbent assays (ELISAs; R&D Systems, Minneapolis, MN, USA) for matrix metallopeptidase-1 (MMP-1), matrix metallopeptidase-9 (MMP-9), tissue inhibitor of matrix metallopeptidase-1 (TIMP-1), and tissue inhibitor of matrix metallopeptidase-2 (TIMP-2). Cytokine quantitation was conducted with a Mulitplex Luminex assay (R&D Systems) for interleukin-1β (IL-1β), interleukin-6 (IL-6), interleukin-8 (IL-8), monocyte chemoattractant protein-1 (MCP-1), and vascular endothelial growth factor (VEGF). Protein values were normalized to the quantity of DNA in each sample (DNeasy Blood & Tissue Kit, Qiagen, Hilden, Germany). In addition to medium samples, biopsies of ES were also collected at 6 h, day 1, day 3, day 5, and day 7 post laser treatment for histologic evaluation (*n* = 3 per time point and group).

### 2.3. ES Grafting to Immunodeficient Mice

Animal care and use observed NIH guidelines and were approved by the Ohio State University Institutional Animal Care and Use Committee (#2017A00000021-R2). An area of skin 1.5 cm × 1.5 cm in size was excised from the flank of each immunodeficient mouse (*n* = 15 per condition; *Foxn1^nu/nu^*, strain code 002019, Jackson Labs, Bar Harbor, ME, USA), leaving intact the panniculus carnosus. At day 10 or 11 of in vitro culture, ES was cut to 1.5 cm × 1.5 cm grafts and grafts were positioned on the wound sites. To reduce graft contraction observed previously [17], 1.7 cm × 1.7 cm silicone stents were cut from CultureWell Silicone Sheet Material (Grace Bio-Labs, Bend, OR) and were sutured onto the normal skin surrounding the wound site. KT Tape^®^ (KT Health, LLC, American Fork, UT, USA) was used to protect the stent but did not cover the graft site. Grafts were then dressed as previously described [17] and assessed daily, with dressing and/or stent materials replaced if damaged. At two weeks post grafting, all dressings and sutures were removed.

Grafts were evaluated at 2 and 4 weeks post grafting. A Tewameter^®^ TM 300 probe (Courage + Khazaka Electronic GmbH, Köln, Germany) was used to measure transepidermal water loss (TEWL) at both time points to assess healing. For the week 2 time point, grafts were open to air after dressing removal for at least 3 h prior to taking TEWL measurements. Normal mouse skin was used as a baseline, determined from measurements from the ungrafted dorsum of 9 mice. Six to eight animals per graft condition were euthanized at each time point for tissue collection. A small dogbone-shaped punch (3 mm width, 10 mm gauge length) was used to create uniform samples for mechanical analysis; samples were stored in phosphate-buffered saline (up to 4 h) until time of testing. Additional biopsies were embedded in OCT compound for cryosectioning, or snap frozen and stored at −80 °C for subsequent molecular analysis.

### 2.4. Mechanical Analysis

Samples were loaded into a tensile tester (TestResources 100R, Shakopee, MN, USA) and tested at a rate of 2 mm/s until failure. The maximum load was established as the greatest load prior to failure. Linear stiffness was determined from the load vs. position plot, with linear regression analysis performed on the linear region immediately after the initial, non-linear (toe) region (y = mx + b, where m = linear stiffness and R^2^ ≥ 0.95). The area under the curve was calculated by summing the definite integrals between the points on the load vs. position plot. All values are reported at the average ± standard deviation.

### 2.5. Histochemical Staining

Samples frozen in OCT were cryosectioned at 7 µm in thickness and Ridged samples were serially sectioned as previously described [17]. Samples collected during the first 7 days of in vitro culture were immunostained with rabbit anti-lactate dehydrogenase (LDH; #ab130923, Abcam, Cambridge, MA, USA), followed by detection with AlexaFluor 594 anti-rabbit secondary antibody (#A21207, Thermo Fisher Scientific), and nuclei were counterstained with DAPI (Thermo Fisher Scientific). ES samples were additionally stained with a pan cytokeratin antibody, AlexaFluor^®^ 488 (Pan-CK; #53-9003-80, Thermo Fisher Scientific) to distinguish keratinocytes from fibroblasts. Fluorescently labeled sections were imaged using an Olympus FV1000 Filter confocal microscope.

Sections from four animals per graft condition at weeks 2 and 4 post grafting, were stained with Hematoxylin & Eosin (H&E) to visualize general tissue structure and with PicroSirius Red (PSR, Electron Microscopy Sciences, Hatfield, PA, USA) to evaluate collagen presence and morphology within the grafts. PSR-stained sections were imaged with a polarized light microscope (Leica DM2500 LED) with constant capture parameters. A 20× objective was used to capture images of 3 non-overlapping regions per sample, through the depth of the graft. ImageJ was used to quantify collagen content in the dermis using a constant brightness for all images. A color threshold was used to determine presence of red (hue <5 and >210), orange (hue 6–25), yellow (hue 26–40), and green (hue 41–110) pixels. Values for total collagen (sum of all pixels within the color threshold) and for mature (red + orange pixels) and immature (yellow + green pixels) collagen were normalized to the area of the dermis in the field of view (FOV) and are reported as average % Pixels in Dermis per FOV ± standard deviation.

Samples from weeks 2 and 4 post grafting were immunostained to evaluate presence of papillary (rabbit anti-netrin 1, NTN1, # ab126729, and mouse anti-podoplanin, PDPN, #ab10288; Abcam) and reticular (rabbit anti-calponin 1, CNN1, # ab46794, and mouse anti-transglutaminase 2, TGM2, # ab2386; Abcam) fibroblasts in the dermis. Additional sections were also immunostained to assess vascularization, either with rabbit anti-von Willebrand Factor (VWF; #ab6994, Abcam) and alpha smooth muscle actin (αSMA; #14-9760-82, Thermo Fisher Scientific) or with rabbit anti-CD31 (#ab28364, Abcam). These antibodies were detected with AlexaFluor^®^ secondary antibodies (Thermo Fisher Scientific) and nuclei counterstained with DAPI. The Mouse-on-Mouse immunodetection kit (Vector Laboratories Inc., Burlingame, CA, USA; Cat# BMK-2202) was used according to the manufacturer’s protocol for immunostaining with primary mouse monoclonal antibodies to reduce non-specific background staining in grafted samples. For quantitative analysis of CD31 staining, 4 samples from each condition were stained (2 non-overlapping FOV/sample) for analysis. ImageJ was used to determine the number of pixels positive for CD31 in the dermis and values are reported as average % Pixels in Dermis per FOV ± standard deviation. The linear regions of CD31 staining observed in the dermis were quantified by counting stained regions consisting of at least two continuous cells and are reported as average number of linear regions per FOV ± standard deviation.

### 2.6. Quantitative Gene Expression Analysis

Biopsies from Flat and Ridged grafts were collected at weeks 2 and 4 post grafting for quantitative PCR (qPCR) analysis of gene expression. Total RNA was isolated using the RNeasy Mini Kit (Qiagen, Inc., Germantown, MD, USA) and was treated with DNase I (Qiagen) prior to preparation of cDNA using the SuperScript VILO cDNA Synthesis Kit (Thermo Fisher Scientific). qPCR was performed using gene-specific RT^2^ qPCR Primer Assays (Qiagen; see Appendix A for product numbers), Power SYBR Green qPCR Mastermix (Thermo Fisher Scientific; Cat# 4368708), and the StepOne Plus Real-Time PCR System (Thermo Fisher Scientific). Expression was analyzed for inflammation-associated proteins IL-6, IL-8, MCP-1, and heat shock proteins HSP47 (SERPIN H1), HSP70 (HSP-A1A), and HSP72 (HSP-A1B). Genes associated with papillary dermal fibroblasts (podoplanin, PDPN) [29] and papillary extracellular matrix (ECM) (decorin, DCN) [30], and reticular dermal fibroblasts (TGM2) [29] and reticular ECM (versican, VCAN) [31,32], were also analyzed. Expression levels were referenced to the glyceraldehyde 3-phosphate dehydrogenase gene (GAPDH, Qiagen) using the comparative 2−∆∆Ct method [33] and relative expression levels were determined by normalizing to mean expression in Flat (control) grafts at week 2. Statistical outliers were determined via Minitab (Minitab, Inc., State College, PA, USA) and the average normalized expression ± standard deviation is reported.

### 2.7. Statistical Analyses

Analysis of data was performed with Minitab. A One-Way Analysis of Variance (ANOVA) with a Tukey post hoc test was used to determine the difference between graft conditions and time points. A *p* value ≤ 0.05 established statistical significance.

## 3. Results

### 3.1. In Vitro Response to Laser Patterning

Dermal templates in the Ridged group were laser-treated to introduce a pattern of microarchitectural features enabling rete ridge formation (Figure 1); non-lasered Flat dermal templates served as controls. Keratinocytes were inoculated onto the surface of Flat dermal templates, or immediately following laser patterning of Ridged dermal templates, to form bilayered ES. LDH immunostaining at 6 h and days 1, 3, 5, and 7 post HK inoculation showed viable fibroblasts and keratinocytes throughout the tissue, including immediately surrounding the ablated areas (Figure 2).

Protein secretion was analyzed between groups over time (Figure 3). IL-β1, IL-6, IL-8, MMP-1, MCP-1, TIMP-1, and TIMP-2 expression followed a general trend of being significantly increased in the range of 6 h to day 3, and then decreasing at days 5 and 7 to levels near or below that of 6 h samples (*p* > 0.05 for both Flat and Ridged grafts, for day 7 vs. 6 h). VEGF followed a similar trend, peaking at day 1 (*p* < 0.001 for Flat grafts, day 1 vs. 6 h and day 5, and *p* < 0.01 for Ridged grafts, day 1 vs. all time points), though secretion for both Ridged and Flat grafts at day 7 was ~2.5 fold higher than at 6 h. Secretion of IL-6, MCP-1, TIMP-1, and TIMP-2 was significantly elevated in Flat samples at some early time points, but showed no significant difference from Ridged samples at days 5 or 7. MMP-9, however, increased in expression over the course of the experiment (*p* < 0.05 for Flat, day 5 and day 7 vs. 6 h and day 1, and *p* < 0.01 for Ridged, day 7 vs. all time points), with secretion in Ridged samples appearing to be slightly delayed compared to Flat samples, although there was no significant difference between groups.

### 3.2. Engineered Skin Graft Assessment and Morphology

Both Ridged and Flat ES were engrafted after transplantation to mice (Figure 4A) with no differences in engraftment between groups. H&E staining revealed a well-developed epidermis for both groups that appeared to densify and become more cellular over time (Figure 4B). The interdigitations of the Ridged grafts likewise densified and became slightly shallower from weeks 2 to 4. The dermis of Ridged samples showed greater cellularity at week 2 than Flat samples, although they were similar at week 4. The dermal ECM also appeared to be more robust in Ridged grafts at both weeks 2 and 4. Contraction, which is inversely proportional to percent of original wound area, significantly increased over time in both groups (*p* < 0.001, Figure 4A,C). At 2 weeks post grafting, Flat grafts had retained approximately 11% more of their initial area than Ridged grafts. However, at week 4, the two groups had contracted to within 1% of each other. Epidermal barrier function, which was assessed by measuring TEWL, was similar between groups and was not significantly different than normal mouse skin at weeks 2 and 4 post grafting (Figure 4D).

### 3.3. Graft Biomechanics

The biomechanics of Flat and Ridged ES after grafting was analyzed using mechanical testing (Figure 5A–C). Both maximum load at failure (Figure 5A) and linear stiffness (Figure 5B), measures of tissue strength, were significantly increased in Ridged samples at both time points, and values were roughly 2-fold higher in comparison to Flat samples. Additionally, the area under the curve, a measure of work or energy expended until failure, was significantly increased in Ridged samples at the early time point (Figure 5C).

### 3.4. Analysis of Collagen Structure

PSR staining of samples post grafting showed much greater collagen presence in the dermis of Ridged samples at both weeks 2 and 4 (Figure 5D). Image quantification revealed significantly more total collagen (Figure 5E) and more mature collagen in Ridged samples (red and orange pixels, Figure 5F) versus Flat samples at both time points. In contrast with Ridged grafts, Flat grafts showed a large amount of inter graft variability at week 4, with half of the analyzed samples being increased ~5-fold in mature collagen versus the other half. Overall, immature collagen was present but not significantly different between groups and total dermal collagen increased from weeks 2 to 4 for both groups (*p* < 0.05).

### 3.5. Analysis of Vascularization and Tissue Development

Vascularization was analyzed by localization of VWF and CD31, which are expressed by endothelial cells, and α-SMA, which is expressed in pericytes of blood vessels as well as in myofibroblasts. Localization of VWF and α-SMA was similar in both Flat and Ridged groups at 2 weeks post grafting (Figure 6A). However, by week 4 there was more VWF and αSMA in Ridged grafts, and noticeably more in the superficial dermis near the DEJ. Quantification of vascularization via CD31 immunostaining (Figure 6B) and image analysis revealed a significant increase in vascular density in Ridged grafts at week 2 post-grafting, both in the percentage of positive-staining area (Figure 6C) and in the number of linear regions of positive staining (Figure 6D). However, by week 4, both metrics had decreased in Ridged grafts and were similar to those of Flat grafts.

Markers of papillary and reticular fibroblasts were localized in sections of ES to further assess tissue development (Appendix A). At week 2 post grafting, Flat grafts showed a mix of both papillary (NTN1 and PDPN) and reticular (TGM2 and CNN1) fibroblast-associated markers. Ridged grafts at this time point, however, were primarily only positive for papillary fibroblast markers. At week 4, papillary markers were found throughout the dermis in both sample groups, with the highest intensities observed for PDPN. Ridged grafts at week 4 showed some positive CNN1 staining, though this was limited to blood vessels.

### 3.6. Analysis of Gene Expression

To determine whether laser treatment induced an inflammatory or heat shock response that might influence tissue development and wound healing, gene expression of major inflammatory cytokines IL-6, IL-8, and MCP-1 were analyzed, along with heat shock proteins HSP-A1A (also known as HSP70), HSP-A1B (also known as HSP-72), and SERPIN-H1 (also known as HSP47). No significant differences in expression of IL-6, IL-8, MCP-1, HSP-A1A, HSP-A1B, or SERPIN-H1 were detected between Flat and Ridged grafts at either in vivo time point (Figure 7). The two groups also generally followed the same trend, except for MCP-1 expression, which increased for Flat grafts and decreased in Ridged grafts over time. Additionally, there were no significant differences in gene expression for PDPN, TGM2, DCN, and VCN between the groups at either time point (Appendix A). Significant differences with time were observed between Ridged grafts for SERPIN H1, HSP-A1A, PDPN, and TGM2.

## 4. Discussion

In this study, the in vitro inflammatory response, and tissue remodeling and mechanics post grafting, were analyzed in ES containing rete ridges compared with Flat, control ES. As previously observed in response to laser micropatterning [17], rete ridge-like structures developed in laser-treated ES in vitro and persisted for 4 weeks post grafting. Prior to this study, it was unclear whether ridge formation was completely driven by the physical change in the dermal construct or if an altered cell phenotype in response to the laser treatment contributed to rete ridge development. The residual heat after lasering, and accompanying alteration to the ECM in which the cells reside, specifically in the microthermal zones (MTZs), may contribute to alterations in cellular phenotype. The extent of MTZs is dependent on laser energy, and therefore, cell phenotype and overall tissue structure can be modified by changes in laser protocol and MTZ size [34]. In previous work, we observed a decrease in F-actin staining in fibroblasts immediately surrounding the ablated wells [17], likely due to the interruption of collagen structure. Here, we observed LDH-positive fibroblasts adjacent to the MTZs, indicating the absence of additional cell death outside of the ablated regions for one week following laser treatment.

To investigate potential alterations in cellular phenotype in response to laser treatment, we analyzed protein secretion of inflammatory cytokines and other factors involved in wound healing and tissue remodeling from ES samples in vitro, and measured gene expression after grafting in vivo. Both control and laser-patterned dermal templates exhibited pro-inflammatory cytokine expression in vitro. IL-6 and IL-8, which were upregulated at early time points, have been found to promote monocyte chemotaxis and induce differentiation into M1 macrophages [35], and MCP-1 has been linked with monocyte chemotaxis, macrophage recruitment [36], and skin fibrosis [37,38]. However, the elevated expression of these cytokines had fallen by day 5 in vitro, before the time of grafting. We hypothesize that the combination of keratinocyte–fibroblast communication and contact inhibition following keratinocyte seeding led ES samples to behave similarly to a re-epithelialized wound and dampened the inflammatory phase of wound healing. A downregulation of TIMP-2 and an upregulation of MMP-9 agrees with this hypothesis that keratinocyte co-localization limits the inflammatory response and pushes wound healing into the remodeling phase [39]. Cultured skin substitutes were previously shown to display increased expression of genes associated with inflammation, angiogenesis, and keratinocyte activation or proliferation, including IL-6 and MMP-9 [40], compared with native human skin, suggesting a wound-healing phenotype that resolves after grafting [41].

FXCO_2_ laser treatment of normal human skin has been shown to result in detection of HSP47 from day 7 to 3 months post treatment [42] and peaking at 1 month [43], HSP70 from 4 h through 1 month post treatment [43], and HSP72 from day 2 through 3 months post treatment [42]. However, our results did not indicate any long-term HSP upregulation as a result of the laser micropatterning, possibly due to the fabrication process. Immediately after laser micropatterning, dermal templates were moved to culture dishes with fresh medium, so any free-floating cell or scaffold debris was most likely rinsed away. Also, laser treatment occurred prior to keratinocyte seeding and, as such, keratinocytes were not directly affected by the laser. In normal human skin, in addition to the dermal regions surrounding MTZs and within the subsequent microlesions, HSP70 presence was highest in the epidermal spinocellular layer and HSP47 was most noticeable in the basal epidermal layer [43]. Here, the cell–cell communication was likely different between the unaffected keratinocytes and residual fibroblasts versus what is typically seen in laser-treated full-thickness skin.

Despite some early differences between Flat and Ridged grafts in cytokine secretion in vitro, no statistically significant differences were seen in cytokines after culture day 5 or in inflammatory gene expression through 4 weeks post grafting. However, the collagen deposition, maximum load, and linear stiffness of Ridged grafts were significantly improved over Flat grafts at both time points post grafting. These results suggest that the morphological change brought about by the laser is the driving force for rete ridge development and tissue remodeling in this construct. One effect of the laser holes could be to create niches that facilitate keratinocyte growth and stratification. The proliferation of keratinocytes within such niches alters local mechanical forces, and cells within the niches experience contact inhibition, which then limits cell division [44,45]. It has previously been observed that deep, narrow channels promote epithelial proliferation and differentiation with increases in keratinocyte stem cell populations found at the tips and edges of the channels [8]. The architecture of the laser patterned dermal template also increases the area of contact between the two cell compartments and enhances cell–cell communication, while also changing force distribution within the tissue.

Another possible effect of the laser treatment that supports rete ridge development could be that it disrupts the fibroblast-seeded collagen network, altering tissue mechanics and possibly decreasing tension in the voids of the dermal template where keratinocytes are seeded. This is consistent with the reduction in f-actin in fibroblasts surrounding MTZs in dermal templates observed in our previous study [17]. Fibroblasts [46,47,48,49] and keratinocytes [46,50,51,52] are known to be sensitive and responsive to their mechanical environment. When cultured on collagen-coated polyacrylamide gels, keratinocytes displayed increased migration velocities and colony-formation rates, with an increased number of cells per colony, when seeded on soft (nominal *E* = 1.2 kPa) versus stiff (nominal *E* = 24 kPa) gels [53]. Softer substrates also appeared to better support cell–cell communication, as keratinocytes generated larger substrate deformations and recruited other local keratinocytes to join colonies to a higher degree on soft gels [53]. The laser-induced change in substrate mechanics may support the increase in keratinocyte proliferation seen previously in Ridged grafts [17]. Additionally, tension has a significant impact on wound healing and remodeling [54] and, with culture time, it is possible that fibroblasts in the Ridged grafts are able to generate more tension in between these voids while keratinocyte growth also generates tension within the epidermal ridges, leading to more rapid tissue remodeling and improved biomechanics. Finally, the accelerated early vascularization observed in Ridged grafts may have contributed to enhanced tissue development by enabling more rapid delivery of nutrients and circulating effector cells to the healing skin grafts.

## 5. Conclusions

Laser micropatterning of fibroblast-populated dermal templates prior to keratinocyte seeding enables the fabrication of ES with rete ridges without inducing a significant, lasting inflammatory response in vitro or in vivo. The results suggest that the development of rete ridges in this ES model is encouraged by the physical removal of ECM and cells, which create microarchitectural features that promote rete ridge formation in vivo and is not related to inflammation. Additionally, this development of rete ridges was shown to significantly improve graft strength and stiffness without hindering barrier function in vivo. Thus, the engineering of skin substitutes with rete ridges enables full-thickness wound closure and results in healed skin tissue with improved biomechanical properties and greater homology with uninjured human skin.

## Figures and Tables

**Figure 1 bioengineering-10-00861-f001:**
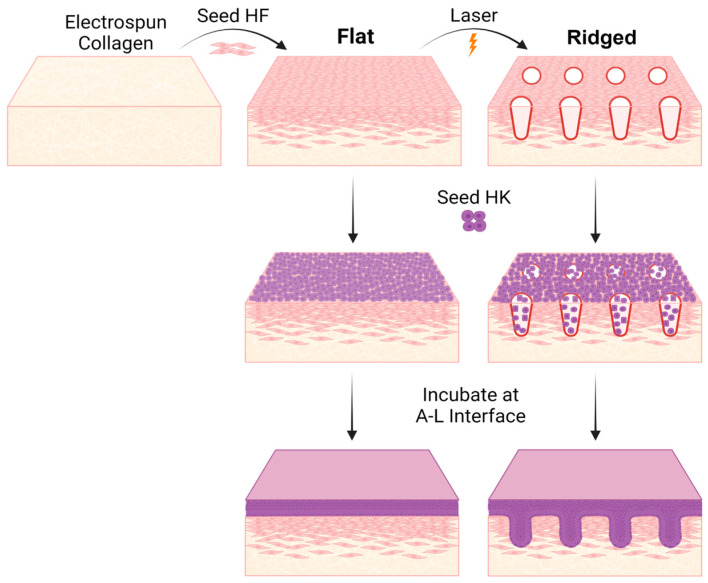
Preparation of Flat and Ridged engineered skin grafts. Human dermal fibroblasts were seeded onto electrospun collagen scaffolds (upper left). After five days of culture, dermal templates were laser micropatterned with a fractional carbon dioxide ablative laser to create rete ridges; control (Flat) dermal templates were not laser-treated. Keratinocytes were seeded immediately after laser treatment and grafts were lifted to the air–liquid (A–L) interface 24 h later to facilitate epidermal stratification. Created with BioRender.com.

**Figure 2 bioengineering-10-00861-f002:**
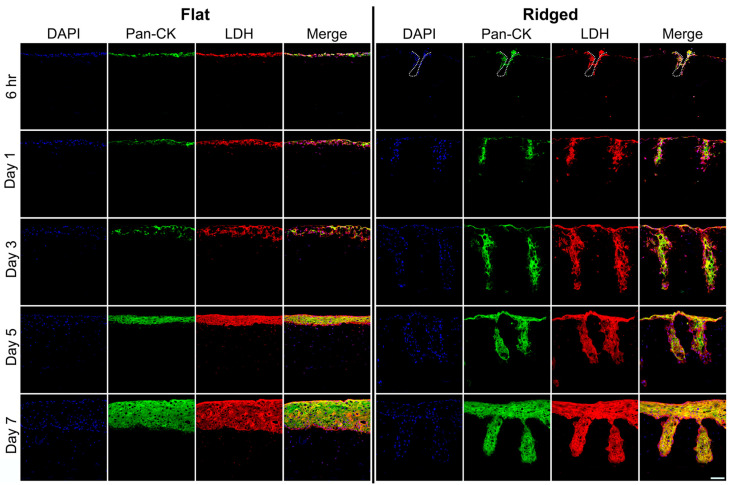
Immunohistochemical localization of pan-cytokeratin (Pan-CK; green) and lactate dehydrogenase (LDH; red) in Flat and Ridged ES at in vitro culture time points 6 h and days 1, 3, 5, and 7 after keratinocyte inoculation. Nuclei were counterstained using DAPI (blue). LDH-positive, viable fibroblasts were found in both groups, including in the microthermal zones of Ridged grafts. Dashed line (top row) indicates boundary of ablation. Scale bar (100 μm, bottom right) is the same for all panels.

**Figure 3 bioengineering-10-00861-f003:**
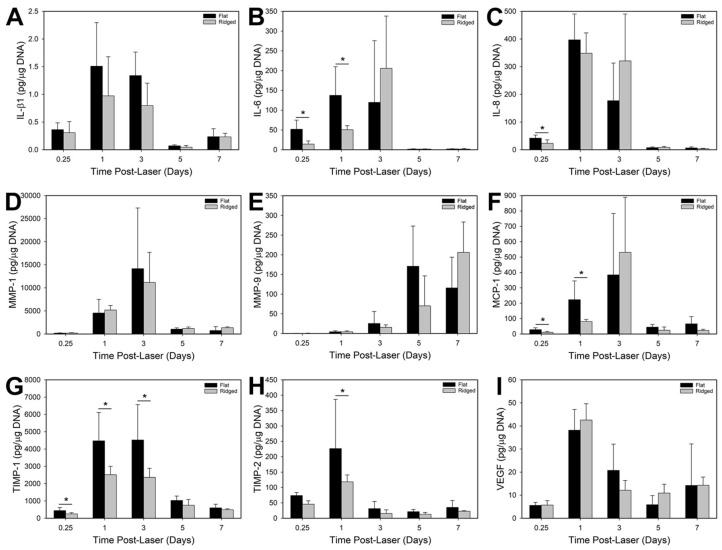
Secretion of inflammatory cytokines and proteins involved in wound healing in ES in vitro. Shown are quantitative analyses of IL-β1 (**A**), IL-6 (**B**), IL-8 (**C**), MMP-1 (**D**), MMP-9 (**E**), MCP-1 (**F**), TIMP-1 (**G**), TIMP-2 (**H**), and VEGF (**I**) secretion from Flat and Ridged ES at in vitro culture time points 6 h (0.25 days) and days 1, 3, 5, and 7 after keratinocyte inoculation. Protein levels were normalized to DNA content for each sample. Protein secretion was similar or decreased for Ridged grafts at early time points and was statistically similar between the two groups at 5 days post laser treatment. Asterisks (*) indicate significant differences (*p* < 0.05).

**Figure 4 bioengineering-10-00861-f004:**
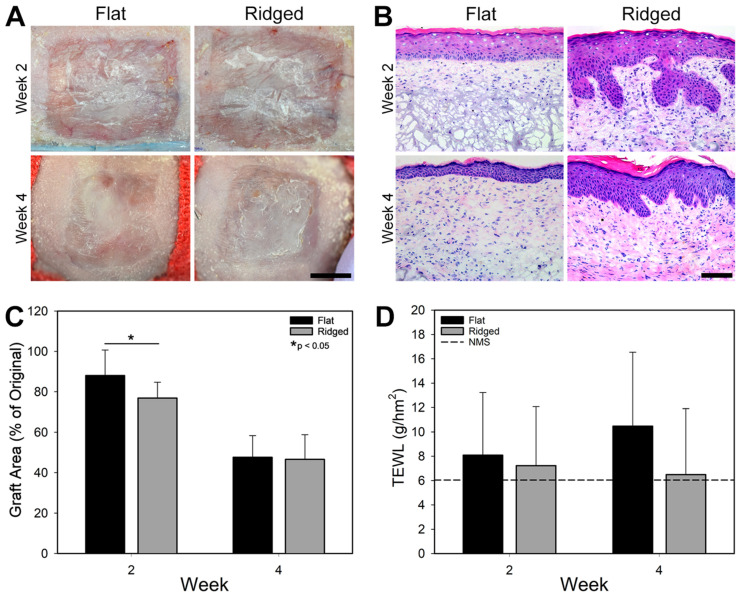
Graft characteristics at weeks 2 and 4 post grafting. (**A**) Macroscopic images of Flat and Ridged ES grafts in vivo, showing similar gross appearance in both groups. Scale bar = 500 μm. (**B**) Representative H&E stained sections of Flat and Ridged grafts at 2 and 4 weeks after grafting, illustrating rete ridge formation at 2 and 4 weeks only in laser micropatterned grafts. Scale bar = 100 μm. (**C**) Graft area, which is inversely proportional to contraction, plotted as a percentage of area at time of grafting. Significantly more contraction was observed in Ridged grafts at 2 weeks post grafting (*, *p* < 0.05), but no significant difference was observed at 4 weeks post grafting. (**D**) No differences in barrier function, assessed by measuring transepidermal water loss (TEWL), were observed between groups at 2 or 4 weeks post grafting.

**Figure 5 bioengineering-10-00861-f005:**
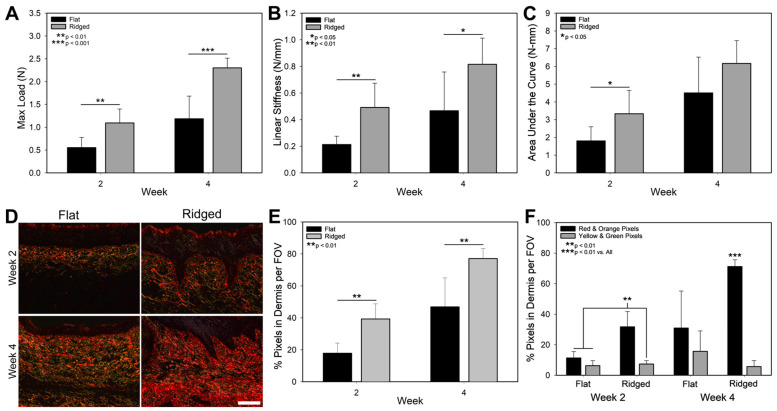
Mechanical analysis and picrosirius red staining of Flat and Ridged grafts at weeks 2 and 4 post grafting. Maximum load at failure (**A**), linear stiffness (**B**), and area under the curve (**C**) were determined from the force-displacement plot of samples tested to failure at a rate of 2 mm/s. Ridged grafts displayed a significant increase in area under the curve at week 2 post grafting, and in maximum load and linear stiffness at both time points. Representative images (**D**) and image analysis of picrosirius red staining for total collagen (**E**) and for mature (red and orange pixels) and immature collagen (yellow and green pixels, (**F**)) showed significant increases in mature collagen and total collagen presence in Ridged grafts at both time points. Scale bar = 100 μm.

**Figure 6 bioengineering-10-00861-f006:**
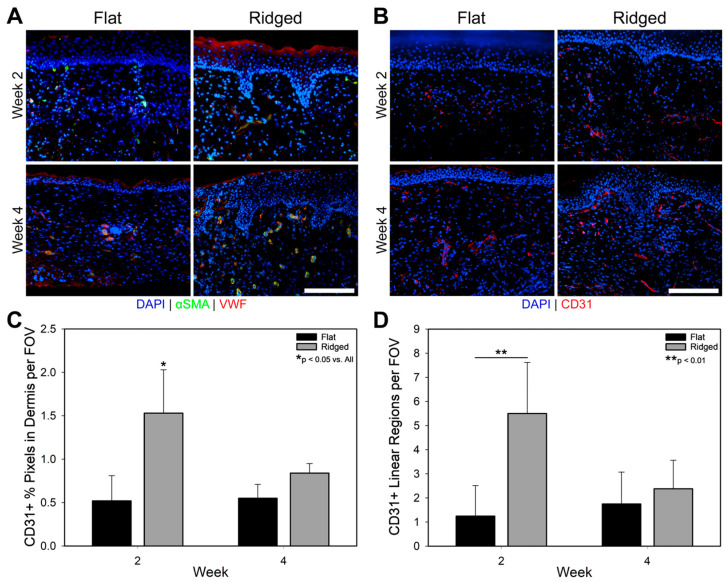
Immunohistochemical (IHC) localization of vascularization in Flat and Ridged grafts at 2 and 4 weeks post grafting. (**A**) IHC staining for alpha smooth muscle actin (αSMA; green) and von Willebrand Factor (VWF; red), with a counterstain for nuclei (DAPI; blue). VWF presence appeared to be similar between the two groups at week 2 and increased in Ridged grafts at week 4, especially in the superficial region of the dermis. (**B**) IHC staining for CD31 (red), with a counterstain for nuclei (DAPI; blue), and quantification of CD31 positive pixels (**C**) and number of linear regions of CD31 positive pixels (**D**) normalized to the area of the dermis per field of view. At week 2, Ridged grafts displayed an increase in CD31 presence as well as in the number of linear regions of CD31. Scale bar = 100 µm for all panels.

**Figure 7 bioengineering-10-00861-f007:**
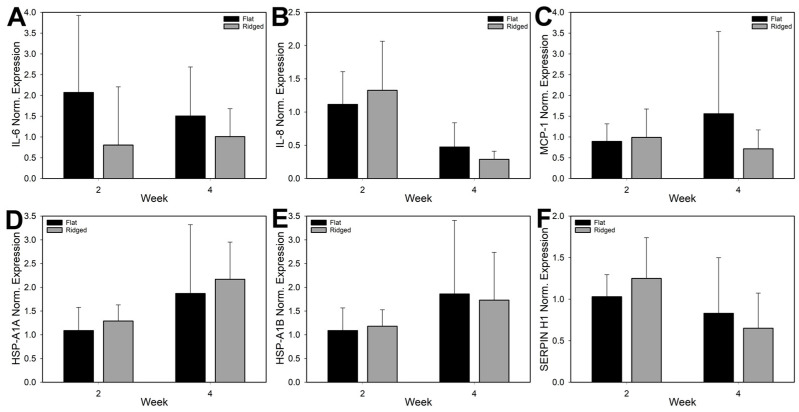
Gene expression analysis for inflammatory-associated proteins IL-6 (**A**), IL-8 (**B**), and MCP-1 (**C**), and heat shock proteins HSP 47 (SERPIN H1) (**D**), HSP70 (HSP-A1A) (**E**), and HSP72 (HSP-A1B) (**F**) in Flat and Ridged grafts at 2 and 4 weeks post grafting. Expression was referenced to GAPDH presence and normalized to mean expression in Flat grafts at week 2. No significant differences were determined between the groups at either time point (*p* > 0.05).

## Data Availability

All relevant data are included in the manuscript. Raw data can be obtained upon request from the corresponding author.

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
