# Peer review of "Laser Micropatterning Promotes Rete Ridge Formation and Enhanced Engineered Skin Strength without Increased Inflammation"

_bioengineering, 2023, doi:10.3390/bioengineering10070861_

Round 1

Reviewer 1 Report

The authors provided a qualitative study, which can be recommended for publication after correcting some shortcomings. 

Remarks that are of a fundamental and that should be made include:

The used cells "Fibroblasts and keratinocytes isolated from de-identified human skin acquired from the breast reduction of a 28-year-old African 119 American female" must be at least characterized or have references if it was ealry charcterized.

For a better and more visual understanding of the research results, it is necessary to provide data/classical histological analysis (haematoxylin/eosin staining). Сlassical immunohistochemical staining of ColI/III, aSMA and silver denovo collagen staining would greatly enhance the value of the data presented.

Optional, but desirable requirements include the desire to see the analysis of secretomes, and not selective factors.

As a technical note, the studies did not indicate a large number of catalog numbers used.

Author Response

The authors provided a qualitative study, which can be recommended for publication after correcting some shortcomings. 

Remarks that are of a fundamental and that should be made include:

The used cells "Fibroblasts and keratinocytes isolated from de-identified human skin acquired from the breast reduction of a 28-year-old African 119 American female" must be at least characterized or have references if it was ealry charcterized.

-Thank you for your comment. We have amended the text to include more information about the cell characterization.

For a better and more visual understanding of the research results, it is necessary to provide data/classical histological analysis (haematoxylin/eosin staining). Сlassical immunohistochemical staining of ColI/III, aSMA and silver denovo collagen staining would greatly enhance the value of the data presented.

-H&E stained sections were shown in Figure 4. Additionally, collagen I/III staining (picrosirius red) was shown in Figure 5D.

Optional, but desirable requirements include the desire to see the analysis of secretomes, and not selective factors.
Thank you for your suggestion. At this time, quantifying a large portion of the secretome is out of the scope of this work. 

As a technical note, the studies did not indicate a large number of catalog numbers used.
Catalog numbers were added.

Reviewer 2 Report

Comments to the authors:

1. Fig 7 is missed in the main text. 

2. "Analysis of Gene Expression" is not clear, which factors/genes expression have effective impact on wound healing process.

3. Conclusion doesn't meet the requirements.

Minor editing of English language required. 

Author Response

  1. Fig 7 is missed in the main text. 

         Thank you.  Figure 7 has been included in the resubmission.

  1. "Analysis of Gene Expression" is not clear, which factors/genes expression have effective impact on wound healing process.
    Additional text was added to support the reason for choosing analyze the genes selected.
  2. Conclusion doesn't meet the requirements.
    There were no requirements listed for the conclusion (the conclusion was actually listed as optional). Could the reviewer clarify their question?